# Biodiversity and Abundance of Cultured Microfungi from the Permanently Ice-Covered Lake Fryxell, Antarctica

**DOI:** 10.3390/life8030037

**Published:** 2018-09-06

**Authors:** Laurie Connell, Benjamin Segee, Regina Redman, Russell J. Rodriguez, Hubert Staudigel

**Affiliations:** 1School of Marine Sciences and Molecular and Biomedical Sciences, University of Maine, Orono, ME 04469, USA; ben.e.segee@maine.edu; 2Adaptive Symbiotic Technologies, University of Washington, Seattle, WA 98195, USA; rsredman@adsymtech.com (R.R.); rjrodriguez@adsymtech.com (R.J.R.); 3Scripps Institute of Oceanography, La Jolla, CA 92037, USA; hstaudig@ucsd.edu

**Keywords:** Antarctic, fungi, Lake Fryxell, permanently ice-covered lakes, extreme habitats

## Abstract

In this work, we explore the biodiversity of culturable microfungi from the water column of a permanently ice-covered lake in Taylor Valley, Antarctica from austral field seasons in 2003, 2008 and 2010, as well as from glacial stream input (2010). The results revealed that there was a sharp decline in total culturable fungal abundance between 9 and 11 m lake depth with a concurrent shift in diversity. A total of 29 species were identified from all three water sources with near even distribution between Ascomycota and Basidomycota (15 and 14 respectively). The most abundant taxa isolated from Lake Fryxell in 2008 were *Glaciozyma watsonii* (59%) followed by *Penicillium* spp. (10%), both of which were restricted to 9 m and above. Although seven species were found below the chemocline of 11 m in 2008, their abundance comprised only 10% of the total culturable fungi. The taxa of isolates collected from glacial source input streams had little overlap with those found in Lake Fryxell. The results highlight the spatial discontinuities of fungal populations that can occur within connected oligotrophic aquatic habitats.

## 1. Introduction

Lake Fryxell in Taylor Valley, geographically part of the McMurdo Dry Valleys region of Antarctica is a permanently ice-covered lake with an average of 6 m of ice. This area is the focus of decades of research and more recently tourism, therefore may be vulnerable to human impacts. This lake is primarily fed by melt water from the Canada and Commonwealth Glaciers (Figure 1—photo of area with glaciers feeding lake). A marked chemocline exists at 11 m with increases in both salts and SO_4_^2−^ concentrations [1]. An active microbial loop exists in Lake Fryxell as described by Bowman and coworkers [2] is based on decades of microbial research on prokaryotes [3,4,5,6,7,8,9,10] and eukaryotes [7,11,12,13,14,15,16,17,18]. However, as with most lakes worldwide, the study of fungal biodiversity and ecological significance lags far behind [19]. One of the earliest studies of fungi associated with Lake Fryxell [20] described culturable yeasts in the algal mats that form on the lake shore but, sampling did not include either lake water or sediment. 

The uniqueness of the McMurdo Dry Valleys ice covered lakes and the recent human activity, both research and tourist, highlights the importance of assessing the status of and potential human impact on these microbial communities. Here we describe cultivable fungal species isolated from Lake Fryxell water at several depths below a permanent ice cover as well as from streams at the terminus of the two major glaciers feeding Lake Fryxell during austral field seasons 2003, 2008 and 2010.

## 2. Materials and Methods 

### 2.1. Field Location

Lake Fryxell is located in Taylor Valley, McMurdo Dry Valleys, South Victoria Land, Antarctica (−77.62; 163.15). The glacial streams were sampled within 5 m of the glacial front of Canada Glacier (−77.72; 163.11), and Commonwealth Glacier (−77.69; 163.41) (Figure 1).

### 2.2. Sample Collection and Processing

Water samples were collected from Lake Fryxell through a hole drilled in the lake ice during 2003, 2008 and 2010 using a 5-L Niskin bottle. All sampling was done during the Austral summer, specifically during the months of December and January. The water depth was determined from the level that the lake water raised in the sample hole. Samples were collected at 7, 8 and 9.5 m during 2003; at 7, 8, 9, 11 and 12 m during 2008; and at 7, 8, 9, 10, 11, 12, 13 and 14 m as well as glacier terminus streams during 2010. Stream samples were obtained by collecting 10 L of running stream water in sterile 1 L bottles. The sample size for all samples, regardless of source, was considered to be 5 L. Each 5-L sample was filtered onto Metricel black (Gelman) 47 mm 0.45 sterile membrane filters that were then incubated on media plates of YPD (BD, Franklin Lakes, NJ, USA) or 50% YPD each containing chloramphenicol (100 mg^−1^·mL) at 10 °C. The 2008 samples were filtered in 100 mL aliquots with all filters plated separately and all emerging isolates were enumerated. For the other field seasons some water samples clogged the filters with 100-mL volumes and those samples were filtered with smaller volumes per filter and each complete water sample was utilized. Each filter was plated separately and at least five of each morphotype, if there were five on a plate, were selected from each plate for further processing.

Culture plates were shipped refrigerated from Antarctica to University of Maine and incubated at 10 °C for up to one year. Each culture plate was checked for fungal growth and photographed at weekly interval for up to 18 months from time of collection. 

Multiple isolates of each observed morphotype were subcultured from 2003 and 2010 samples. The number of isolates investigated varied based on the number of morphotypes observed. If available at least five of each morphotype were identified. A subculture from every isolate obtained in 2008 was generated and identified using either F-ARISA (fungal-automated ribosomal intergenic spacer analysis) as previously described [21] or by DNA sequencing of the ribosomal internal transcribed spacer region (ITS) for those isolates that could not be positively identified using F-ARISA. The F-ARISA is based on the variability of ITS fragment sizes. In this method, polymerase chain reaction (PCR) is carried out and the resultant fragments are compared using high resolution capillary DNA fingerprinting. Multiple isolates from cultures acquired in 2003, 2010 and from stream samples were subcultured and identified by sequencing the ITS region. Sets of primer combinations including ITS4, ITS5, ITS1F, EF3 [22,23,24,25] were used for PCR amplification. PCR for DNA sequenced fragments were carried out as previously described [21]. Specific ITS region amplicons were produced by PCR completed with 100 ng genomic DNA in 25 μL reactions using Illustra PuReTaq Ready-To-Go™ PCR Beads (GE Lifesciences, Pittsburgh, USA). PCR primer set ITS5-ITS4 (White et al. 1990) were used to target the ITS region for sequencing (Table 1). Initial denaturation completed for 2 min at 95 °C and 35 cycles with a PTC-200 thermal cycler (MJ Research, Watertown, MA, USA) under the following conditions: 30 s at 95 °C, 30 s at 52.3 °C, 1 min at 72 °C with a final 72 °C 10 min extension. DNA extraction and F-ARISA from all cultured 2008 isolates, regardless of origin, was carried out as previously described [26] and at least one isolate from each taxa identified was sequenced, even for those identified using the F-ARISA method. DNA sequencing was done by the University of Maine Sequencing facility (https://umaine.edu/dnaseq/). The internal transcribed spacer (ITS) sequences, including ITS1 through ITS2, were aligned using MUSCLE [27]. The resulting alignment was used to create a phylogenetic tree through the Seaview version 4.3.1 [28]. A rooted neighbor-joining distance tree was generated using the Seaview option of Jukes–Cantor distance measure [28]. Bootstrap values were based on 100 replicates. Venn’s diagram was made using Venny 2.1 [29]. ITS sequences were submitted to GenBank [21] and are shown in Table 1 for all identified taxa.

## 3. Results

### 3.1. Fungal Diversity

Thirty species of fungi were cultured from Lake Fryxell waters and glacial streams feeding Lake Fryxell (Table 1, Figure 2). There was about equal representation between Ascomycota and Basidiomycota species found from all the sources (53% to 47% respectively). 

During 2008 we were able to determine the culture-based abundance of fungal taxa identified from Lake Fryxell. The 2008 Lake Fryxell samples showed *Glaciozyma* species representing 69% of the total abundance and *Toxicocladosporidium*, *Acremonium*, and *Geomyces* each representing 1% (Figure 3).

### 3.2. Fungal Distribution

The distribution of fungi was explored both by depth in Lake Fryxell as well as potential seeding by input by glacial streams. Isolates from all three field seasons were used for distribution determination. The total fungal abundance above the sharp chemocline of 11 m was higher than below with little over all abundance difference between 7 m and 9 m (Figure 4 and Figure 5, Table 1). Although *Glaciozyma* spp. and *Penicillium* spp. comprised the largest number of the fungal abundance above 11 m, they were not found in samples below 11 m. There were several species found both above and below the 11-m chemocline boundary, however only three species were found exclusively below 11 m (*Clavispora lusitaniae, Holtermanniella nyarrowii*, and *Toxicocladosporium strelitziae*).

Potential fungal seeding of Lake Fryxell by stream input was explored by sampling stream water near the fronts of the two primary glaciers that feed the lake. A total of 13 species were identified from Canada Glacier stream and Commonwealth Glacier stream (Table 1). There was no overlap of species found between either of these sources (Figure 5). In addition, only five of these 13 species were found in Lake Fryxell samples, while 17 species found in lake water were not found in either of the glacial stream samples.

## 4. Discussion

Although few studies have examined fungal diversity in Antarctic lake water columns, there have been studies that have isolated fungi from soil surrounding lakes, lake sediment, or nearby by microbial mats [30]. For example, lake sediment samples from the Skarvsnes Oasis region of East Antarctica revealed a diversity of five Ascomycetes and five Basidiomycetes genera similar to that found in Lake Fryxell [30]. Even over such a large geographic distance some of the same genera (*Mrakia* & *Thelebolus*) were isolated from the Skarvsnes Oasi lake sediments and Lake Fryxell (the Skarvsnes Oasi study characterized isolates only to genus). This suggests that these genera are cold adapted and common across Antarctica. More intriguing are the isolates from Skarvsnes Oasi soils that overlap with Lake Fryxell isolates (e.g., *Geomyces*, *Cryptococcus*, *Mrakia*, *Theolobus*, *Diozegia*). The fact that glaciers and stream do not appear to be a major source of fungi to the lake, the overlap between the studies suggest that soil may be a significant source of fungi in Lake Fryxell. Fungi may move from the soil to the lake by wind dispersion, melt pond run-off, or direct movement via snowmelt or growth.

Lake Fryxell has increasing H_2_S and decreasing dissolved O_2_ below the 11-m chemocline [31]. The abundance of only a few viable fungal species that are found primarily below the chemocline suggests that these organisms are not only able to tolerate the local environment but may be adapted to it. This is supported by the fact that, although most fungi are inhibited by H_2_S, a few species have been reported to metabolize and grow at high levels of H_2_S [32,33]. Additional research is needed to determine if these fungi are actively metabolizing and/or growing below the chemocline. There have been several studies of the microbial composition in Lake Fryxell, focusing on bacteria and *Archaea* [4,5,12,14,34,35]. Methanogenic Archaeal diversity [4] in Lake Fryxell were found to be clustered in either the water column or the sediment, demonstrating functional separation. In addition, a diversity of phototropic purple bacteria was identified from the water column [5,36] and nutrient cycling both above and below the chemocline. 

The fact that there are at least four sources of microbial introduction into Lake Fryxell (e.g., CG, CWG, soil, and air) suggests that either: (1) The spatial distribution of source fungi is extreme and there is little or no homogenization of microbes over this area. (2) The level of detection may be so low that the error rates are high creating a detection bias from the small sample sizes that are used in these kinds of studies. (3) Soil may be a significant source of fungi as indicated above.

The list of fungi found in Antarctica continues to grow with the suggestion that “fungi might be the most diverse biota in Antarctica” [37]. There remain many challenges to determine both fungal abundance and activity from Antarctic environmental samples yet is clear that they must play a role in nutrient recycling and decomposition. Based on the extreme habitat below the Lake Fryxell chemocline, it is clear that a limited number of fungal species are able to adapt yet it remains to be determined how closely the Lake Fryxell isolates are to those found in the northern hemisphere.

## Figures and Tables

**Figure 1 life-08-00037-f001:**
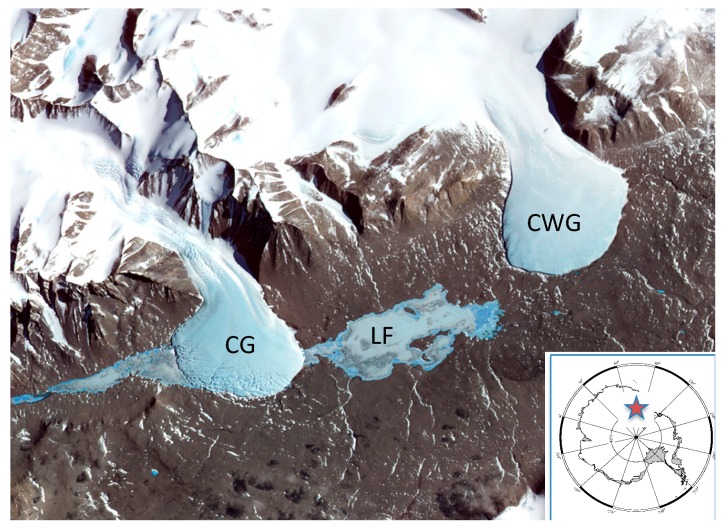
Taylor Valley Antarctica showing Lake Fryxell (LF) between Canada (CG) and Commonwealth Glaciers (CWG). The location of Taylor Valley is shown by the star on the insert of Antarctica.

**Figure 2 life-08-00037-f002:**
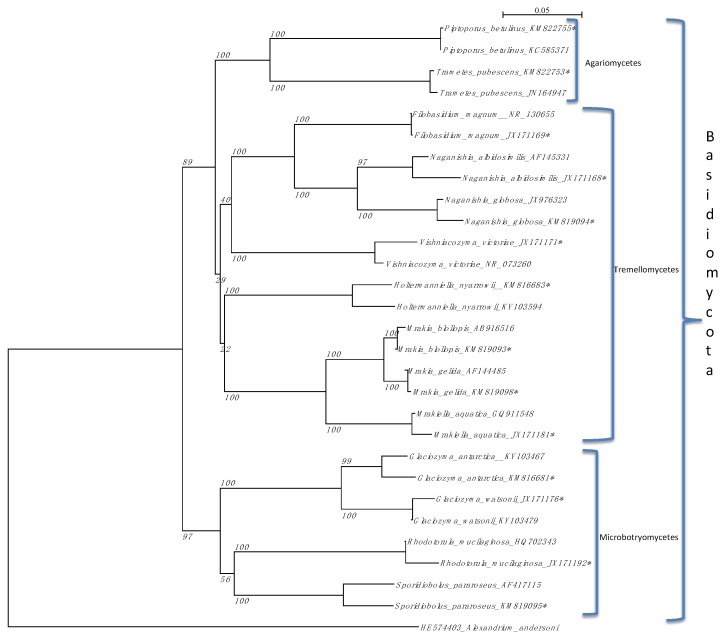
Neighbor Joining tree of the ITS region of rDNA to illustrate the relationship of sequences obtained from yeasts isolated from Lake Fryxell as well as Commonwealth Glacier and Canada Glacier streams as a simple cladogram. Isolates from this work are identified by asterisk (*). Other isolates shown are from GenBank. The outgroups for both Ascomycetes groups (**upper**) and Basidiomycetes (**lower**) were the alga *Alexandrium andersoni*.

**Figure 3 life-08-00037-f003:**
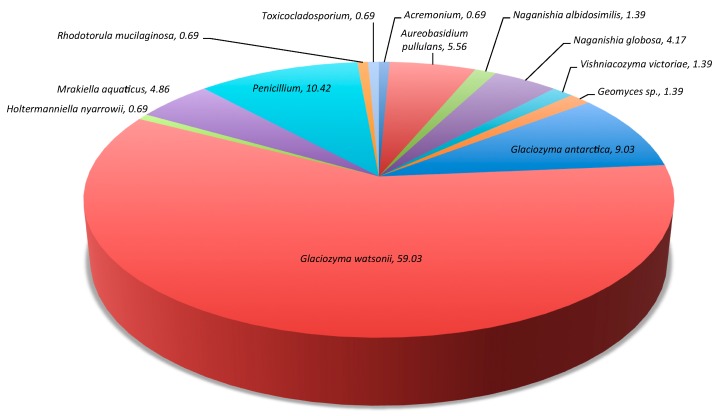
Percent of abundance for each fungal taxa identified from Lake Fryxell during 2008.

**Figure 4 life-08-00037-f004:**
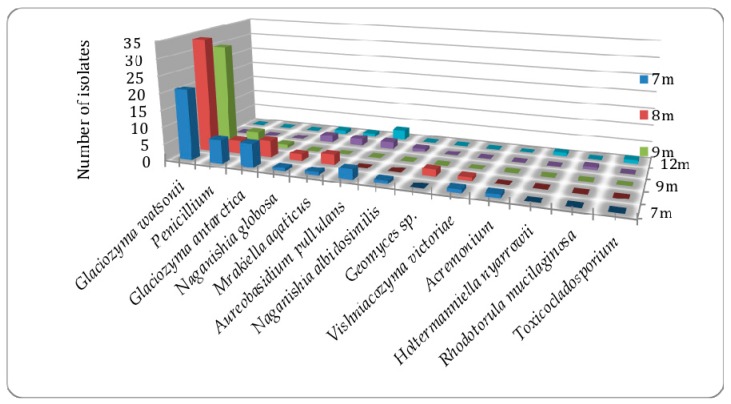
Abundance of cultured fungi in Lake Fryxell by depth in meters from the 2008 field season.

**Figure 5 life-08-00037-f005:**
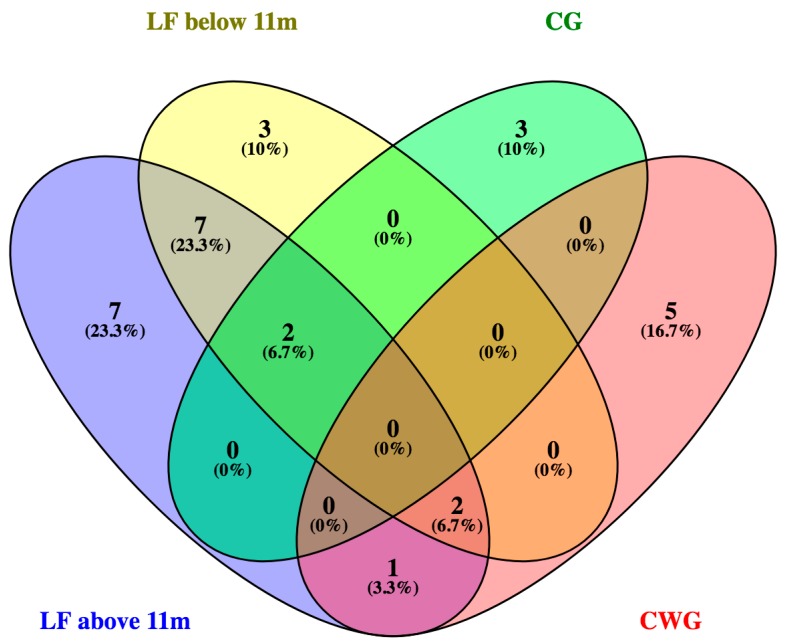
A Venn’s diagram of fungal taxa shown by Lake Fryxell (LF) isolate above and below 11 m depth as well as streams feeding Lake Fryxell of Canada Glacier stream (CG) and Commonwealth Glacier stream (CWG).

**Table 1 life-08-00037-t001:** Species identified from Lake Fryxell (LF), Canada Glacier stream (CG), and Commonwealth Glacier stream (CWG).

Species	GenBank Accession #	A^1^	B^2^	LF above 11 m	LF below 11 m	CG	CWG
*Filobasidium magnus*	JX171169		X	X	X		
*Debaryomyces hansenii*	KM816678	X		X			
*Thelebolus ellipsoideus*	JX171195	X		X	X	X	
*Thelebolus globosus*	JX171196	X		X	X		X
*Geomyces sp. 1*	KM816679	X		X			
*Acremonium sp.*	KM816680	X		X			
*Mrakiella aquatica*	JX171181		X	X	X		
*Rhodotorula mucilaginosa*	JX171192		X	X	X		X
*Vishniacozyma victoriae*	JX171171		X	X			
*Glaciozyma antarctica*	KM816681		X	X			
*Geomyces sp. 2*	KM816682	X		X	X	X	
*Glaciozyma watsonii*	JX171176		X	X			
*Naganishia globosa*	KM819094		X	X	X		
*Naganishia albidosimilis*	JX171168		X	X	X		
*Penicillium dipodomyicola*	JX171186	X		X			
*Holtermanniella nyarrowii*	KM816683		X		X		
*Aureobasidium pullulans*	JX171163	X		X	X		
*Toxicocladosporium strelitziae*	KM816684	X			X		
*Cladosporium cladosporoides*	KM816685	X		X	X		
*Penicillium commune*	JX171184	X		X			X
*Heydenia alpina*	JX171178	X		X	X		
*Clavispora lusitaniae*	KM816686	X			X		
*Mrakia gelida*	KM819098		X				X
*Mrakia blollopsis*	KM819093		X			X	
*Sporidiobolus pararoseus*	KM819095		X			X	
*Cladosporium sp.*	KM819097	X				X	
*Trichoderma atrovirde*	KM822752	X					X
*Trametes pubescens*	KM822753		X				X
*Eutypa lata*	KM822754	X					X
*Piptoporus betulinus*	KM822755		X				X

A^1^ = Ascomycota, B^2^ = Basidiomycota.

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
