# Peer review of "Biodiversity and Abundance of Cultured Microfungi from the Permanently Ice-Covered Lake Fryxell, Antarctica"

_life, 2018, doi:10.3390/life8030037_

Round 1
Reviewer 1 Report
Report of manuscript: Life_318877
Title: Biodiversity and abundance of cultured microfungi from the permanently ice-covered Lake Fryxell, Antarctica
Authors: Laurie Connell, Benjamin Segee, Regina Redman, Russell Rodriguez, and Hubert Staudigel.
Recommendation: Minor review
In this work, the authors carry out the identification of several fungal species from different samples collected at a lake located in Antarctica. They cultured fungal isolates and identified the species through F-ARISA or ITS sequencing. The manuscript is well written and in my opinion suitable for publication in Life. I only have some minor comments.
-Abstract, line 27: A sentence summarizing the importance and implications of this work.
-Keywords: More keywords should be added.
-Line 35: Add the charge of the sulfate anion
-Line 37: “…decades of on microbial…”. Is this sentence correct?
-Lines 38-39 and lines 43-44: That´s is something that should be discussed but it is not. The authors should include in the Discussion the implications and importance of this work.
-Line 56: Separate the title of this section from the text.
-Did the authors analyze different replicates of the samples? How many?
-Lines 74-76: The authors should show the alignment and the phylogenetic tree,
-Line 79: “ITS sequence” or “ITS sequences”?
-Pictures 2, 3 and 4 could be grouped in one single figure with different panels.
-Lines 100-103: Again, any implication for this observation? The authors mention something in the discussion but should elaborate on this.
-Lines 144-147: As potential “extremophiles”, how would the authors analyze how these organisms have adapted to this environment and their potential usefulness?
Author Response
Recommendation: Minor review
In this work, the authors carry out the identification of several fungal species from different samples collected at a lake located in Antarctica. They cultured fungal isolates and identified the species through F-ARISA or ITS sequencing. The manuscript is well written and in my opinion suitable for publication in Life. I only have some minor comments.
-Abstract, line 27: A sentence summarizing the importance and implications of this work.
The sentence “The results highlight the spatial discontinuities of fungal populations that can occur within connected oligotrophic aquatic habitats.” was added
-Keywords: More keywords should be added. Additional key words added”
“permanently ice-covered lakes; extreme habitats”
-Line 35: Add the charge of the sulfate anion This has been done
-Line 37: “…decades of on microbial…”. Is this sentence correct? Corrected to read “based on decades of microbial research”
-Lines 38-39 and lines 43-44: That´s is something that should be discussed but it is not. The authors should include in the Discussion the implications and importance of this work.
This is now part of extensively re-written discussion
-Line 56: Separate the title of this section from the text. This has been done
-Did the authors analyze different replicates of the samples? How many? Text added “Multiple isolates of each observed morph types were subcultured from 2003 and 2010 samples. The number of isolates investigated varied based on the number of morphotypes observed. If available at least five of each morphotype were identified.” In addition, EVERY isolate from 2008 was identified.
-Lines 74-76: The authors should show the alignment and the phylogenetic tree, Two phylogenetic trees have been added.
-Line 79: “ITS sequence” or “ITS sequences”? This has been changed to “sequences”
-Pictures 2, 3 and 4 could be grouped in one single figure with different panels. FIG 4 is different.
The figures were separated because of context. Fig 3 is distribution by taxa and figure 4 is by location. We did keep these separate.
-Lines 100-103: Again, any implication for this observation? The authors mention something in the discussion but should elaborate on this. The discussion has been re-written to include these observations.
-Lines 144-147: As potential “extremophiles”, how would the authors analyze how these organisms have adapted to this environment and their potential usefulness?
Not all of these organisms are extremophiles, many can be found worldwide. Therefore we did make a mention I the re-written discussion, however we could not make extensive conclusions based on data known about the organisms.
Reviewer 2 Report
-It is not clear how fungi from years other than 2008 were identified. Why are the 2008 samples the only ones mentioned in the abstract? The sampling section needs to be reworked. It is very confusing as to what the sampling effort was (see below) per site and what samples were used to characterize each community.
-Are sample sizes the same across depths and sampling sites? In this case, it would be the number of filters plated. I assume this is 100 samples per site per depth (100 mL per sample from 10 Liters = 100). If this is correct or wrong, either way it needs to be clarified. If the sample sizes differ, there needs to be a correction for differences in sample sizes when comparing communities (rarefaction). Looking again, it seems that sampling effort differs between LF and glacial streams so corrections need to be made to determine if abundance differs significantly.
-More explanation is needed in the methods. The authors rely too heavily on referencing other publications.
-Briefly explain the methods for F-ARISA even if referencing another publication.
-What primers are used for amplification? What are the cycling conditions? Where was the sequencing done and how?
-Provide citations for software. Both Muscle and Seaview are published. Cite those publications.
-Did the aligned regions exclude 5.8s? Clarify how the alignment was constructed if certain regions were excluded.
-Genera should be italicized.
-Taxa is plural, taxon is singular. Correct throughout (e.g. caption for figure 2).
-How did the authors choose which fungi to include in Fig. 3? There are 22 fungi from LF in Table 1, but only 13 included in Fig. 3.
-Why does Fig. 4 show 23 taxa from LF while Table 1 shows 22?
-In the discussion, the authors posit the presence of a few species below the chemocline suggests they are active there. I do not see how there is any data presented suggesting they are active. They were removed from this environment then plated. There is no indication of metabolism at the depth at which they were recovered. Fungal spores can often tolerate very adverse conditions. The authors need to further justify this point.
Specific comments:
Line 33: venerable to vulnerable
35: exist to exists
36: parenthetical statement is confusing, please fix grammar for clarity
49: during what?
72: This sentence does not make sense: "DNA extraction and sequencing from all cultured isolates,
regardless of origin, was carried out as previously described [22] and at least one isolate from each
taxa identified was sequenced, even for those identified using the F-ARISA method."
Either everything was sequenced or at least one isolate per taxon was sequenced. Which is it? Clarification is needed here.
125: What do the authors mean by "broad diversity"? Representing diverse lineages within each phylum? Clarify.
Author Response
-It is not clear how fungi from years other than 2008 were identified. Why are the 2008 samples the only ones mentioned in the abstract? The sampling section needs to be reworked. It is very confusing as to what the sampling effort was (see below) per site and what samples were used to characterize each community.
The Abstract was rewritten to reflect all of the years sampled.
-Are sample sizes the same across depths and sampling sites? In this case, it would be the number of filters plated. I assume this is 100 samples per site per depth (100 mL per sample from 10 Liters = 100). If this is correct or wrong, either way it needs to be clarified. If the sample sizes differ, there needs to be a correction for differences in sample sizes when comparing communities (rarefaction). Looking again, it seems that sampling effort differs between LF and glacial streams so corrections need to be made to determine if abundance differs significantly.
Text was added to clarify “The sample size for all samples, regardless of source, was considered to be 5L. Each 5L sample was filtered onto Metricel black (Gelman) 47mm 0.45m sterile membrane filters and incubated on media plates of YPD (BD, Franklin Lakes, NJ) or 50% YPD each containing chloramphenicol (100 mg-1 ml) at 10oC. Some water samples were placed onto a number of plates because filters clogged with larger volumes. These multiple filters from the original 5L samples were considered one sample.”
-More explanation is needed in the methods. The authors rely too heavily on referencing other publications.
Text was added to read:
Multiple isolates of each observed morphotype were subcultured from 2003 and 2010 samples. The number of isolates investigated varied based on the number of morphotypes observed. If available at least five of each morphotype were identified. A subculture from every isolate obtained in 2008 was generated and identified using either F-ARISA as previously described [21] or by DNA sequencing of the ribosomal internal transcribed spacer region (ITS) for those isolates that could not be positively identified using F-ARISA (fungal- automated ribosomal intergenic spacer analysis). The F-ARISA is based on the variability of ITS fragment sizes. In this method PCR is carried out and the resultant fragments are compared using high resolution capillary DNA fingerprinting. Multiple isolates from cultures acquired in 2003, 2010 and from stream samples were subcultured and identified by sequencing the ITS region. Sets of primer combinations including ITS4, ITS5, ITS1F, EF3 [22-25] were used for PCR amplification. DNA extraction and F-ARISA from all cultured 2008 isolates, regardless of origin, was carried out as previously described [26] and at least one isolate from each taxa identified was sequenced, even for those identified using the F-ARISA method. DNA sequencing was done by the University of Maine Sequencing facility (https://umaine.edu/dnaseq/). The ITS sequences, including ITS1 through ITS2, were aligned using MUSCLE [27]. The resulting alignment was used to create a phylogenetic tree through the Seaview software program, version 4.3.1 [28]. A rooted neighbor-joining distance tree was generated, based on 251 nucleotide positions of the ITS 1 and 2 regions, using the Jukes-Cantor distance measure. Bootstrap values were based on 100 replicates. Venn’s diagram was made using Venny 2.1 [29]. ITS sequences were submitted to GenBank [21] and are shown in Table 1 for all identified taxa.
-Briefly explain the methods for F-ARISA even if referencing another publication.
Text was added
“…….identified using F-ARISA (fungal- automated ribosomal intergenic spacer analysis). The F-ARISA is based on the variability of ITS fragment sizes. In this method PCR is carried out and the resultant fragments are compared using high resolution capillary DNA fingerprinting.”
-What primers are used for amplification? What are the cycling conditions? Where was the sequencing done and how?
Text was added- however the details would be best left for a citation.
Sets of primer combinations including ITS4, ITS5, ITS1F, EF3 [22-25] were used for PCR amplification. PCR for DNA sequenced fragments were carred out as previously described [21]. Specific ITS region amplicons were produced by PCR completed with 100 ng genomic DNA in 25ul reactions using Illustra PuReTaq Ready-To-GoTM PCR Beads (GE Lifesciences). PCR primer set ITS5-ITS4 (White et al. 1990) were used to target the ITS region for sequencing (Table I). Initial denaturation completed for 2min at 95oC and 35 cycles with a PTC-200 thermal cycler (MJ Research, Watertown, MA) under the following conditions: 30 sec at 95oC, 30 sec at 52.3oC, 1 min at 72oC with a final 72oC 10 min extension.
-Provide citations for software. Both Muscle and Seaview are published. Cite those publications.
Those citations have been added
-Did the aligned regions exclude 5.8s? Clarify how the alignment was constructed if certain regions were excluded.
Text was edited to “DNA ITS sequences, including ITS1 through ITS2, were aligned…”
-Genera should be italicized.
This has been done
-Taxa is plural, taxon is singular. Correct throughout (e.g. caption for figure 2).
This has been done
-How did the authors choose which fungi to include in Fig. 3?
The fungi in Fig 3 and 4 are from 2008. This was the only year we were able to determine abundance based on culturing. The other field seasons only unique morphotypes were selected but not counted.
-Why does Fig. 4 show 23 taxa from LF while Table 1 shows 22?
Figure 4 (now figure 5) has been edited
-In the discussion, the authors posit the presence of a few species below the chemocline suggests they are active there. I do not see how there is any data presented suggesting they are active. They were removed from this environment then plated. There is no indication of metabolism at the depth at which they were recovered. Fungal spores can often tolerate very adverse conditions. The authors need to further justify this point.
The discussion has been extensively re-written
Specific comments:
Line 33: venerable to vulnerable
This has been done
35: exist to exists
This has been done
36: parenthetical statement is confusing, please fix grammar for clarity
This has been changed
49: during what?
This has been added
72: This sentence does not make sense: "DNA extraction and sequencing from all cultured isolates,
Re-written to read “regardless of origin, was carried out as previously described [22] and at least one isolate from each taxa identified was sequenced, even for those identified using the F-ARISA method."
Either everything was sequenced or at least one isolate per taxon was sequenced. Which is it? Clarification is needed here.
Re-written to read “DNA extraction and F-ARISA from all cultured 2008 isolates, regardless of origin, was carried out as previously described [22] and at least one isolate from each taxa identified was sequenced, even for those identified using the F-ARISA method.”
125: What do the authors mean by "broad diversity"? Representing diverse lineages within each phylum? Clarify.
The word broad was eliminated and the sentence re-written to read “For example, lake sediment samples from the Skarvsnes Oasis region of East Antarctica revealed a diversity of five Ascomycetes and five Basidiomycetes genera similar to that found in Lake Fryxell [30].”
Reviewer 3 Report
Manuscript: “Biodiversity and abundance of cultured microfungi from the permanently ice-covered Lake Fryxell, Antarctica.
Merits:
Conventional culture based microbiology analysis is a rarity these days. Author’s effort to execute a culture-based experiment is highly appreciated.
Improvements:
· The reviewer feels that this manuscript needs a thorough revisit to be considered for publication. Mentioned below are few places where the author can further improve.
· Line 31- 35: Rewrite the sentences to make the message clear for the reader. For eg.
Start with “Lake Fryxell in Taylor Valley, geographically part of the McMurdo Dry Valleys region of Antarctica is …………….”.
· Line 39 – 41: Rewrite the sentence
· Line 42: Remove "Because of the"
· Line 56 – 61: Information being given by the author is not clear
· Line 61 – 65: Rewrite the paragraph, language is not scientific friendly.
· Use standard naming convention for microbial species “italicize” all genus and species name throughout the paper
· Use the “Cultivable fungi species abundance” as term “abundance” is contemporarily used in respect to the metagenome studies.
· Table 1: Please mention the sampling time/year with the species, cumulatively presenting the data does not give a clear understating of the microbial shift.
· Line 95: reads "Fungal distribution" and the author starts with "Fungal diversity" which has already been discussed in the previous paragraph.
· Figure 3: “italicize” all genus and species name
· Even though sampling was done during 2003, 2008 and 2010, why has author only discuss 2008 field sample only? A comparative co-relative analysis will make more ecological valued data.
· The reviewer suggests that discussion should be rewritten and subjective terms should be used in right context.
· Line 130: change “no” to “not”
· Line 131 – 135: Context of the paragraph is not clear,
· Line 136 – 140: Author has not given any supporting evidence for this part of the discussion. Rewrite and explain properly
· Line 144 – 147: Reviewer feels that Metagenomics study will give a better picture of abundance and adaptation, the cultivable study should not be used as a definitive answer to that.
General Comments:
There is a lot of scope for improvement in scientific writing; the author should also have a thorough look at grammar and spacing.
Author Response
· Line 31- 35: Rewrite the sentences to make the message clear for the reader. For eg.
Start with “Lake Fryxell in Taylor Valley, geographically part of the McMurdo Dry Valleys region of Antarctica is …………….”.
Changed as suggested “Lake Fryxell in Taylor Valley, geographically part of the McMurdo Dry Valleys region of Antarctica is a permanently ice covered lake with an average of 6m of ice..”
· Line 39 – 41: Rewrite the sentence ????
· Line 42: Remove "Because of the"
This has been done
· Line 56 – 61: Information being given by the author is not clear
Changed to “Lake Fryxell is located in….” and “The glacial streams were sampled within 5m….”
· Line 61 – 65: Rewrite the paragraph, language is not scientific friendly.
This has been change to “Water samples were collected from Lake Fryxell through a hole drilled in the lake ice during 2003, 2008 and 2010 using a 5-liter Niskin bottle. The water depth was determined from the level that the lake water raised in the sample hole. Samples were collected at 7, 8, and 9.5m during 2003; at 7, 8, 9, 11, and 12m during 2008; and at 7, 8, 9, 10, 11, 12, 13 and 14m as well as glacier terminus streams during 2010. Stream samples were obtained by collecting 10L of running stream water in sterile 1L bottles. The sample size for all samples, regardless of source, was considered to be 5L. Each 5L sample was filtered onto Metricel black (Gelman) 47mm 0.45m sterile membrane filters and incubated on media plates of YPD (BD, Franklin Lakes, NJ) or 50% YPD each containing chloramphenicol (100 mg-1 ml) at 10oC. Some water samples were placed onto a number of plates because filters clogged with larger volumes. These multiple filters from the original 5L samples were considered one sample. Culture plates were shipped refrigerated from Antarctica to University of Maine and incubated at 10oC for up to one year. Each culture plate was checked for fungal growth at weekly intervals.”
· Use standard naming convention for microbial species “italicize” all genus and species name throughout the paper
This has been done
· Use the “Cultivable fungi species abundance” as term “abundance” is contemporarily used in respect to the metagenome studies.
This has been changed
· Table 1: Please mention the sampling time/year with the species, cumulatively presenting the data does not give a clear understating of the microbial shift.
All the sampling was done during the Austral summer and now included in the Methods section to read
“Water samples were collected from Lake Fryxell through a hole drilled in the lake ice during 2003, 2008 and 2010 using a 5-liter Niskin bottle. All sampling was done during the Austral summer, specifically during the months of December and January. The water depth was determined from the level that the lake water raised in the sample hole. Samples were collected at 7, 8, and 9.5m during 2003; at 7, 8, 9, 11, and 12m during 2008; and at 7, 8, 9, 10, 11, 12, 13 and 14m as well as glacier terminus streams during 2010. Stream samples were obtained by collecting 10L of running stream water in sterile 1L bottles. The sample size for all samples, regardless of source, was considered to be 5L. Each 5L sample was filtered onto Metricel black (Gelman) 47mm 0.45m sterile membrane filters and incubated on media plates of YPD (BD, Franklin Lakes, NJ) or 50% YPD each containing chloramphenicol (100 mg-1 ml) at 10oC. Some water samples were placed onto a number of plates because filters clogged with larger volumes. These multiple filters from the original 5L samples were considered one sample. Culture plates were shipped refrigerated from Antarctica to University of Maine and incubated at 10oC for up to one year. Each culture plate was checked for fungal growth at weekly intervals.”
· Line 95: reads "Fungal distribution" and the author starts with "Fungal diversity" which has already been discussed in the previous paragraph.
This has been changed to read “The distribution of fungi”
· Figure 3: “italicize” all genus and species name
This has been done
· Even though sampling was done during 2003, 2008 and 2010, why has author only discuss 2008 field sample only? A comparative co-relative analysis will make more ecological valued data.
We discuss the total fungal diversity from all three seasons throughout. The study did not collect total cultured fungal abundance during the first and last field seasons in this manuscript because of funding limitations. Therefore an analysis comparing various seasons would be of limited valueText has been added to the discussion .
· The reviewer suggests that discussion should be rewritten and subjective terms should be used in right context.
The discussion has been extensively rewritten and expanded.
· Line 130: change “no” to “not”
This has been done
· Line 131 – 135: Context of the paragraph is not clear.
Text has been added for clarily “Even over such a large geographic distance some of the same genera (Mrakia & Thelebolus ) were isolated from the Skarvsnes Oasi lake sediments and Lake Fryxell (the Skarvsnes Oasi study characterized isolates only to genus). This suggests that these genera are cold adapted and common across Antarctica. More intriguing are the isolates from Skarvsnes Oasi soils that overlap with Lake Fryxell isolates (e.g. Geomyces, Cryptococcus, Mrakia, Theolobus, Diozegia). The fact that glaciers and stream do not appear to be a major source of fungi to the lake, the overlap between the studies suggest that soil may be a significant source of fungi in Lake Fryxell. Fungi may move from the soil to the lake by wind dispersion, melt pond run-off, or direct movement via snowmelt or growth.”
· Line 136 – 140: Author has not given any supporting evidence for this part of the discussion. Rewrite and explain properly
The discussion has been extensively expanded and rewritten
· Line 144 – 147: Reviewer feels that Metagenomics study will give a better picture of abundance and adaptation, the cultivable study should not be used as a definitive answer to that.
It may be true that metagenomics would give a better picture of adaptation, however when this work was being carried out metagenomics were expensive and not in the research budget.
General Comments:
There is a lot of scope for improvement in scientific writing; the author should also have a thorough look at grammar and spacing.
Round 2
Reviewer 2 Report
The authors have largely addressed my concerns, but there are a few things they should consider correcting.
While they have characterized a single sample as a 5L water sample, the actual sampling effort could vary among individual water collections due to the number of filters plated. I would contend that the sample unit is actually a single filter. One could pass 5L thru a single filter and plate that filter and should not expect the number of fungi recovered to be the same as if 100 mL were passed through 50 filters (5L total) and those filters were plated. Thus, the number of filters per water sample matters. The impact and utility of this work to others is limited without this information.
Line 94: Jukes-Cantor is a model of sequence evolution, not a distance measure. A model of sequence evolution is used to correct for multiple hidden substitutions that would not be accounted for by some distance measures (ex. uncorrected p-distance). What distance measure was used?
Figure 2 does not display a "Sequence analysis". It is a neighbor joining tree.
Author Response
responses are below each comment-
While they have characterized a single sample as a 5L water sample, the actual sampling effort could vary among individual water collections due to the number of filters plated. I would contend that the sample unit is actually a single filter. One could pass 5L thru a single filter and plate that filter and should not expect the number of fungi recovered to be the same as if 100 mL were passed through 50 filters (5L total) and those filters were plated. Thus, the number of filters per water sample matters. The impact and utility of this work to others is limited without this information
the text has been changed to clarify the amount of samples used per filter.
" Each 5L sample was filtered onto Metricel black (Gelman) 47mm 0.45 sterile membrane filters that were then incubated on media plates of YPD (BD, Franklin Lakes, NJ) or 50% YPD each containing chloramphenicol (100 mg-1 ml) at 10oC. The 2008 samples were filtered in 100mL aliquots with all filters plated separately and all emerging isolates were enumerated. For the other field seasons some water samples clogged the filters with 100mL volumes and those samples were filtered with smaller volumes per filter and utilize each complete water sample. Each filter was plated separately and at least five of each morphotype, if there were five on a plate, were selected from each plate for further processing."
Line 94: Jukes-Cantor is a model of sequence evolution, not a distance measure. A model of sequence evolution is used to correct for multiple hidden substitutions that would not be accounted for by some distance measures (ex. uncorrected p-distance). What distance measure was used?
As stated in the methods section we used the Seaview software for tree building. below is part of the instruction manual that lists J-C as a choice for distance.
User tree: computes the number of steps of a user-given tree taken from those in the "Trees" menu. Such trees can have been previously computed or imported from an external source in Newick format.
Distance methods:
Computes NJ or BioNJ trees on a variety of pairwise phylogenetic distances.
NJ/BioNJ: to select the tree-building algorithm
Save to file: does not compute any tree but saves sequence pairwise distances to a local file.
Distance: select one among a variety of evolutionary distances: J-C: Jukes & Cantor (1969); K2P: Kimura (1980) JME 16:111; HKY: Rzhetsky & Nei (1995) MBE 12:131; LogDet: Lake (1994) PNAS 91:1455; Lockhart et al. (1994) MBE 11:605; Ka/Ks: Li (1993) JME 36:96.
WE did make an alteration to the text to read: "A rooted neighbor-joining distance tree was generated, based on 251 nucleotide positions of the ITS 1 and 2 regions, using the Seaview option of Jukes-Cantor distance measure [28]."
Figure 2 does not display a "Sequence analysis". It is a neighbor joining tree.
This has been changed in the figure legend